



# Induced Electromagnetic prospecting for the characterization of the European southernmost glacier: the Calderone Glacier, Apennines, Italy.

Mirko Pavoni[1], Jacopo Boaga[1], Alberto Carrera[2], Stefano Urbini[3], Federico de Blasi[4,5] and Jacopo Gabrieli[4,5]

[1] Department of Geosciences, University of Padova, Padua, Italy
[2] Department of Agronomy, Food, Natural Resources, Animals and Environment, University of Padova, Legnaro (PD), Italy
[3] Istituto Nazionale di Geofisica e Vulcanologia, Dipartimento Ambiente, Rome, Italy
[4] Institute of Polar Sciences, National Research Council (CNR-ISP), Venice, Italy [5] Department of Environmental Sciences, Informatics and Statistics, Ca' Foscari University of Venice, Venice, Italy

*Correspondence to*: Mirko Pavoni (mirko.pavoni@phd.unipd.it)

**Abstract.** The increasing rate of glacier retreat in recent decades is well documented and represents a great loss for the paleoclimate studies. In this framework, Ice Memory project aims to extract and analyze ice cores from worldwide glacier regions and then storage them in Antarctica as heritage for future generations. Ice coring projects usually require a focused geophysical investigation, often based on the Ground Penetrating Radar (GPR) technique and the active seismic prospection, in order to assess the most suitable drilling positions. As novelty, in the Calderone Glacier, we integrated the GPR results with a Frequency Domain Electro-Magnetic (FDEM) prospection which is not commonly applied in the glacial environment. A separated-coils FDEM instrument has been used to characterize the glacier up to several tens of meters of depth. The acquired FDEM datasets were inverted and compared to the GPR data and borehole information. The results demonstrate the ability of the FDEM instrument to correctly define the structure of the glacier and therefore its potential to be applied in frozen subsoils studies. All this opens new perspectives for the use of FDEM technique to characterize glacial or periglacial environments as rock glaciers, where the GPR acquisition logistic is limited by the rock blocky surface and affected by the scattering from surface debris.

**Keypoints**: FDEM, EMI, GPR, Calderone Glacier, Cryosphere, Environmental Geophysics

## 1 Introduction

The Calderone Glacier is the southernmost ice body in Europe and the last one in the Apennines mountains (Pecci et al., 1997). It develops within the massif of the Gran Sasso d'Italia (Abruzzo, Central Italy) and, like many other alpine glaciers (Crepaz et al., 2013), it is in a retirement phase since the beginning of the 20th century (Marinelli & Ricci, 1916). This trend, connected to an increase in average annual temperatures (Pecci et al., 2008), had a clear acceleration since the 1960s (Tonini, 1961; Smiraglia & Veggetti, 1991; Gellatly et al., 1992, 1994). Today the massive ice core, which has been estimated to have a maximum thickness of 26 meters in 2015 (Monaco and Scozzafava, 2017), is completely covered by a debris layer of several meters. This downward trend of Alpine and Apennines ice bodies is an important proxy of the climate change rate (Haeberli et al., 2007), but at the same time it represents a serious loss as regards the paleoclimatic studies. In fact, geochemical analyses on the ice samples extracted from the glaciers allow the reconstruction of climate and temperatures tendency of the past (Stenni, 2005). To save this important natural database, the international project 'Ice Memory' has been created. The main focus of this project, recognized by UNESCO, is to collect and to store ice samples from glaciers that could disappear or dramatically retire in the next future due to global warming. The extracted ice cores will be finally moved into Antarctica where they will compose a precious paleoclimatic archive accessible to future generations of scientists. Since 2016, the international Ice



memory team has collected ice cores from seven glaciers around the worldwide glacier regions. High-altitude glacier field
campaigns were carried out in Europe, South America and Asia. In the Andes, Caucasus end Tibetan plateau, the ice cores
were extracted respectively from Illimani, Elbrus and Belukha glaciers. In the Alps, ice samples were collected on Col du
dome, Corbassiere and Gorner glaciers. Recently, the Italian Ice Memory team (composed by the Institute of Polar Science of
the Italian national council of research ISP-CNR and Ca' Foscari University of Venice) has planned to extract an ice core from
the last remaining ice body in the Apennines, the Calderone Glacier.
The localization of a meaningful ice core position is the first challenge of each drilling campaign. For this reason, preliminary
geophysical investigations are applied in order to define the main morphologies under the ice, its thickness and its internal
layering status. The GPR method is historically and commonly used with success in glacier environment prospections (Arcone
et al., 1995; Maurer & Hauck 2007; Forte et al., 2015; Church et al., 2021). Pure ice has a relatively low dielectric constant
which doesn't attenuate the high-frequency electromagnetic signal (in the order of MHz) transmitted by the instrument. The
thickness of the ice layer can be precisely estimated since the interface with the underlying bedrock (which on the contrary has
a relatively high dielectric constant) is highlighted by a clear reflection in the acquired radargram (Urbini et al. 2010, 2019).
In the Calderone Glacier, the coring operation was scheduled in the end of April 2022, while the preliminary geophysical
surveys were planned in the middle of March 2022. The presence of several meters of snow cover didn't allow to apply Electric
Resistivity Tomography (ERT) and active seismic methods during the geophysical surveys. Under these circumstances, we
chose to couple the reliable GPR technique with the electro-magnetic prospecting in the frequency domain (FDEM), a
geophysical method rarely applied in glacier environments. The choice was done considering the good results obtained with
the FDEM technique in several alpine rock glaciers and mountain permafrost sites (Boaga et al. 2019; Pavoni et al. 2021).
Thus, on the Calderone Glacier, GPR and FDEM data were acquired along two common lines of investigation, one longitudinal
and one orthogonal to the development of the glacier. Here we compare the results of the two techniques, testing the potential
of the FDEM method in glacial environments. Due to requested depth of investigation, we adopted a separated-coils FDEM
instrument (CMD-DUO, GF-Instruments). Thanks to relatively low frequency of the transmitted signal and wide separations
of the coils, the device was able to reach the bottom of the ice body. The inverted electrical conductivity sections, after an
adequate data filtering, were calibrated with the results of the forward modeling procedure. This was performed considering a
priori information about the different layers that compose the glacier. The obtained results agree with the glacier structure as
suggested by the GPR models and confirmed by the borehole realized on April 31st 2022. In fact, the boundary between the
ice layer and the bedrock was practically found at the same depth predicted by both the geophysical models. In the following
chapters, a description of the survey site and the most recent evolution of the Calderone Glacier will be presented. We introduce
the applied methods (data acquisition and processing) and the results of the investigation. Finally, conclusions and future
development of the work are discussed.
**2 Site description**
The Calderone Glacier is located in Abruzzo (Central Italy – blue circle in Fig.1A), within the massif of the Gran Sasso d'Italia.
It develops at an altitude between 2650-2850 meters above the sea level, on the northern slope of the Corno Grande peak, the
highest summit of the Apennines (2912 m a.s.l.). The Corno Grande is composed entirely of a calcareous succession of the
Triassic platform (Pecci and Mugnozza, 2006). Since the summer of 2000, it has been split into two different ice bodies which
are classified as glacierets (see Fig.1A), i.e. specific snow and ice structures with no downward movement in the last twenty
years. The glacier was able to survive below the limit of perennial snow since it is preserved between steep walls within a
circus facing North-East (Tonini, 1961). Furthermore, the northeastern exposition and the steep rock walls allow to intercept
the winter precipitation coming from eastern Europe so ensuring a very important avalanche feeding. Finally, the ice body is
entirely covered by meters of calcareous debris which acts as a thermal insulator, protecting the ice under-layers from direct



solar radiation and reducing the summer melting. Therefore, the Calderone ice body is now classified as a debris-covered
glacier (Monaco and Scozzafava, 2015) and probably is in a transition phase to a periglacial form (e.g. rock glacier).
Radiometric dating techniques have been performed on the glacial deposits, downstream and on the threshold of the Calderone
circus, confirming that during the Holocene the glacier had various phases of expansion and retreat (Giraudi, 2002). According
to these measurements, the last phase of expansion took place during the Little Ice Age, while the retreat phase is well
documented since the early 1900s. Marinelli & Ricci (1916) estimated that Calderone Glacier covered an area of 0.07 km$^2$ at
the beginning of the 20th century. Tonini (1961) defined its reduction to 0.06 km$^2$ in the 1960s, and in 1990 the surface
decreased by further 20% (Smiraglia & Veggetti, 1992). The glacier was already almost entirely covered by debris in the 90's,
leading to the classification of debris-covered glacier (Gellatly et al., 1992). In March 2022 GPR survey lines (e.g.Fig.1B) and
FDEM investigation lines (e.g. Fig.1C) were acquired to define the point where the ice layer has its maximum thickness.
Among these lines, two have been measured with both the geophysical techniques, Line 1 (green line in Fig.1A) and Line 2
(red line in Fig.1A). The first is longitudinal to the development of the glacier, practically the same orientation of those
performed by Pecci et al. (2001) and Monaco & Scozzafava (2015), while the second one is orthogonal.
**3 Methods**
**3.1 Ground Penetrating Radar (GPR)**
A glacial environment represents a very suitable context for GPR applications since the dielectric properties of ice and snow
lead to a low attenuation of the transmitted signal (Arcone et al., 1995). In the Calderone Glacier survey, GPR measurements
were collected on the snow cover using a GSSI Sir4000 instrument equipped with a 200 MHz digital antenna (see Fig.1B).
Table 1 shows the main acquisition parameters of the GPR survey. All the measurements were georeferred with a Trimble R9s
GPS receiver in RTK configuration. Reflection arrival times were converted in depth using an averaged electro-magnetic wave
speed of 0.201 m/ns and 0.1682 m/ns for the snow cover and the ice layer, respectively. These values have been calculated
by an average of hyperbola diffractions where the medium separations emerged clearly. Data processing, performed uisng
ReflexW software (Sandmeier geophysical research), included the common application of vertical and horizontal bandpass
filters, deconvolution, gain equalization, and migration.
**3.2 Frequency Domain Electro-Magnetic (FDEM) Method**
The FDEM method applies Maxwell's equations to estimate the electrical conductivity of the investigated subsoil (McLachlan
et al., 2021), without the need for a galvanic contact between the device and the ground surface. FDEM instruments have a
transmitter coil ($Tx$) where an alternating current flow with a fixed frequency $f$, inducing a primary magnetic field ($Hp$) with
the same frequency $f$. $Hp$ propagates in the subsoil and induces secondary electrical currents (Boaga, 2017). The latter in turn
generates a secondary electromagnetic field ($Hs$) which is measured by the receiver coil ($Rx$). The ratio between $Hs/Hp$ is a
complex number and from its real part ($Q$) the apparent electrical conductivity ($\sigma_a$) of the subsoil can be calculated, as shown
in Eq.1:
$$Q_a = \frac{4}{\omega \mu_0 s} Q \qquad \text{Eq. 1}$$
where $\omega$ is the angular frequency ($\omega = 2\pi f$) of the transmitted signal, $s$ is the separation of the two coils ($Tx$ and $Rx$), and $\mu_0$ is
the magnetic permeability of free space (considering that most of the subsoils are practically non-magnetic, McLachlan et al.,
2021). This relationship is true only if the Low Induction Number ($\beta$) condition (LIN) is verified:

$$\beta = s\sqrt{\frac{2}{w\mu_0\sigma}} \lll 1 \qquad \text{Eq.2}$$



In a debris covered glacier environment, as the Calderone Glacier, the electrical conductivities are particularly low, and
consequently the LIN condition is always verified. However, the measured $\sigma_a$ is influenced by the contribution of the different
layers that compose the ground. The penetration depth of the measurements is linked to different factors: the separation *s* of
the coils, their orientation (horizontal HCP or vertical VCP), and the transmitted frequency *f*. By using higher coil separations
*s*, the measured apparent conductivity $\sigma_a$ will be more affected by the electrical properties of the deeper layers in the subsoil,
in the same way as using lower frequencies *f*. Finally, considering a fixed value of *s* and *f*, the HCP mode allows to further
increase the penetration depth of the survey respect to the VCP mode (see Fig. 2). In a debris covered environment, with very
low values of electrical conductivities, the magnetic field decays rapidly, restricting the penetration depth (Hauck & Kneisell,
2008). This problem can be partially solved by using a lower frequency *f* and higher values of *s* (Boaga et al., 2020).
Considering these limitations, in the Calderone Glacier we adopted a separated coils FDEM instrument, the GF Instruments
CMD-DUO (see Fig.1C). The device has a low transmitted frequency *f* of 925 Hz, and three relatively large coil separation *s*
of 10, 20, and 40 meters. Moreover, both VCP and HCP modes can be acquired. This way, six $\sigma_a$ values can be obtained in
each measured point (which is considered the halfway between the two coils), defining an electrical conductivity profile from
few meters of depth till several dozens of meters. Fig. 2 shows the nominal depth range, suggested by the manufacturer (GF
Instruments), influencing the measured apparent conductivities acquired with a CMD-DUO device.
The application of the FDEM method in the glacier environment is limited by the instrumental limit resolution, that usually
cannot estimate conductivity below 1E-1 mS/m. The ice of a temperate glacier has an electrical conductivity in the range of
1E-3 mS/m (Hauck & Kneisell, 2008), two orders of magnitude lower than common FDEM instrumental limit. Despite this,
FDEM methods proved to be efficiently applicable in high resistive environments, considering in a relative way the inverted
conductibility profile (e.g. Boaga et al. 2020; Pavoni et al. 2021).
**3.2.1 FDEM forward and inverse modelling**
The forward and inversion FDEM modelling have been performed with the open-source python-based software EMagPy
(McLachlan et al., 2021). To simulate a non-simplified response of the CMD-DUO survey, the Full Maxwell Solution (FS -
Wait, 1982) has been used. The method considers the propagation of electromagnetic fields by conduction currents, valid only
with frequencies $f < 10^5$ Hz (CMD-DUO has a transmitted signal of 925 Hz). The forward modelling consists in the
computation of the Q component of the ratio *Hs/Hp* (eq.3 and eq.4), once considered the characteristic of coil separation *s*,
frequency *f* of the transmitted signal, and the given values of thickness and electrical conductivities of a layered subsoil model:
$$\left(\frac{H_S}{H_P}\right)_{VCP} = 1 - s^2 \int_0^\infty R_0 J_1(s\lambda)\lambda d\lambda \qquad \text{Eq. 3}$$
$$\left(\frac{H_S}{H_P}\right)_{HCP} = 1 - s^3 \int_0^\infty R_0 J_0(s\lambda)\lambda^2 d\lambda \qquad \text{Eq. 4}$$
where $J_0$ is a Bessel function of zeroth order, $J_1$ is a Bessel function of first order, and $R_0$ is the reflection factor, which is
calculated using the thickness and electrical conductivities of the layers (for details see McLachlan et al., 2021). Eq.1 allows
to find a synthetic dataset of $\sigma_a$ that would be measured by the FDEM device, once defined the synthetic subsoil model.
EMagPy was also adopted to perform the quasi-2D inversions of the datasets, generating inverted conductivity profiles in each
measured point. The inverted profiles have been interpolated with the kriging method (Goovaerts, 1997), obtaining a pseudo-
2D conductivity section (from now on simply called as inverted conductivity sections or FDEM models). As for all geophysical
method, the inversion procedure is an iterative process where the software minimizes the misfit between the measured dataset
of $\sigma_a$ and the synthetic dataset of $\sigma_a$ calculated with a forward model. Eq.5 shows the L2 norm objective function which is
minimized for each 1D profile:





$$\frac{1}{N}\sum_{i=1}^{N}(d_i - F_i(m))^2 + \alpha(\frac{1}{M}\sum_{j}^{M-1}(\sigma_j - \sigma_{j+1})^2) \rightarrow min \qquad \text{Eq. 5}$$
In Eq.5, $N$ is the number of coil configurations (separations and orientations), $d$ contains the measured dataset of $\sigma_a$, $F(m)$ the
calculated $\sigma_a$ with the model, $M$ is the number of layers in the model, $\sigma_j$ is the conductivity of layer $j$, and α is the regularization
parameter which can be defined with an L-Curve analysis (Hansen et al, 2001). Among several techniques (see McLachlan et
al., 2021), a straightforward solution to minimize Eq.5 is to use the Cumulative Sensitivity (CS) functions and the gradient-
based optimization method of Gauss-Newton. McNeil (1980) proposed the CS functions, shown in Eq.6 and Eq.7, to define
the contribution of the subsoil layers to the measured apparent conductivities. The normalized sensitivities (R) for the two coil
orientations are:
$$R_{VCP}(z) = \sqrt{(4z^2 + 1)} - 2z \qquad \text{Eq. 6}$$
$$R_{HCP}(z) = \frac{1}{\sqrt{(4z^2+1)}} \qquad \text{Eq. 7}$$
where $z$ is the depth normalized by the coil separation $s$. To facilitate the inversion routine, firstly a data filtering has been
applied. In fact, as the datasets have been acquired in challenging conditions walking with snowshoes on a snow cover of
several meters with steep slopes (see Fig.1B and Fig.1C), it was practically impossible to guarantee the perfect coils orientation
and separation during the measurements. All this inevitably led to the acquisition of anomalous measurements in the acquired
datasets. For these reasons a preliminary data filtering has been applied to the measured datasets, starting from a detrend
function. All the measured $\sigma_a$ outside the confidence interval of eq.8 have been deleted (e.g. Fig.3 presents the filtering of
Line 1 dataset collected with a coil separation of 40 meters and the HCP mode).
$$\mu - 2sd < \sigma_a < \mu + 2sd \qquad \text{Eq.8}$$
where $\mu$ is the average $\sigma_a$ of the dataset and $sd$ is the standard deviation. Subsequently, the saved measurements have been
smoothed, interpolating with a polynomial function of $6^{th}$ grade (e.g. see Fig.3C). Finally, as the numerical inversion modelling
allows to find negative inverted conductivity values, which are obviously unrealistic, we defined a lower boundary of zero for
the inverted conductivity model.
**4 Results**
**4.1 GPR**
Fig.4 shows the post processing results of the GPR measurements. In both the profiles, the snow layer is characterized by low
attenuation of the transmitted signal and the boundary with the underlying frozen debris is characterized by a well recognizable
reflection (red dashed line), as same as the limit between the ice layer and the bedrock (blue dashed lines - see also the raw
measurements in FigA1 and Fig.A2 of the Appendix). The maximum ice thickness value (26.4 m - blue arrow in Fig.4A) has
been found along the longitudinal Line 1 at a distance of ≈ 90 m from the profile start. Along the Line 2, the ice thicknesses
do not show large variations and the thickness differences at the cross-points with L1 are practically negligible (<10%). Note
that, an important signal scattering occurs in the eastern part of the profile, suggesting that here the ice layer has a larger
presence of embedded debris respect to the western part.
**4.2 FDEM inversion results**
Fig.5 shows the results of the FDEM inversion procedure applied to the field datasets acquired along Line 1 (Fig.5A) and Line
2 (Fig.5B), respectively. From a structural point of view, the FDEM sections are very similar to their respective GPR models
(see Fig.4A and Fig.4B). In line 1 (Fig. 5A) a clear low conductivity zone is visible from x ≈ 40 to end of the line, with





maximum thickness between x ≈ 90 and x ≈ 100. Higher conductivity zones are visible in the uppermost layer and in the deeper
part. The same three layers structure can be seen also in the result of Line 2 (Fig.5B), with a structure once again very similar
to the one highlighted by the GPR model (Fig. 4B). However, despite the defined structures are practically the same, the
inverted electrical conductivity values are not realistic, as expected considering the instrumental resolution limits. Synthetic
forward modelling was then computed, to verify and calibrate the obtained results.

### 4.3 FDEM forward modelling results

FDEM synthetic forward models, based on a priori information, were calculated to be compared with the real field dataset.
Synthetic datasets were computed and then inverted, considering the information of 2015 and 2022 GPR surveys. Figure 6A
shows the Calderone Glacier longitudinal model as defined by Monaco & Scozzafava in 2015. Note that, in addition to the
layers defined by the model of Monaco & Scozzafava (2015), a top layer of snow has been added since we had measured
several meters of snow cover during our field test. Figure 6B shows the glacier model along the orthogonal Line 2, this time
basing on the GPR surveys of March 2022. These models have been used to perform the forward modelling process and to
calculate the synthetic datasets simulating a FDEM apparatus with the same properties of the CMD-DUO instrument. The
conductivity of each layer has been defined using both literature values and field measurements, as shown in Table 2. The
conductivity of the snow cover has been fixed to 1 mS/m according to the values measured by Pecci et al. (2006) on the
Calderone Glacier. The frozen calcareous debris conductivity (2E-2 mS/m) has been estimated considering the values found
in the calcareous rock glaciers by Pavoni et al. (2021). The ice of a temperate glacier practically acts as an electrical insulator
and can be set at 1E-3 mS/m (Hauck & Kneisell, 2008). Finally, the bedrock conductivity has been evaluated to be 2E-1 mS/m
(Gélis et al., 2010). The synthetic datasets calculated with the forward modeling procedure have been inverted with the same
procedure of the real data (see 3.2.1). Fig.7 shows the synthetic inverted conductivity models calculated for investigation Line
1 (Fig.8A) and Line 2 (Fig.8B). Considering the results shown in Fig.7, we interpret values of 1E-1 mS/m as the ice rich
layer, and values between 1E-1 and 2E-1 mS/m as an ice-debris mixture. Conductivity values higher than 2E-2 mS/m can be
linked to unfrozen debris in the top layers and to bedrock at the bottom of the section. Values close to 1 mS/m may represent
the upper snow cover layer. It can be note that the subsoil structure of the synthetic FDEM results are very similar to the real
dataset ones (see Fig. 5), but the conductivity values.

### 4.4 FDEM Calibration

The synthetic dataset inversion results (Fig.7) were used to calibrate the real dataset inversion sections (Fig.5). The CMD-
DUO device instrumental limit resolution (1E-1 mS/m) is two orders of magnitude lower than the electrical conductivity of
the massive ice (1E-3 mS/m). Therefore, we did not expect to find inverted conductivity values that matched with the synthetic
dataset inversion. Calibration intends to explore if exist a constant correction factor to be applied to the inversion results of the
field datasets, in order to have the same conductivity scale of the synthetic model.
Considering both the result of the GPR survey line 1 (Fig.4A), and the longitudinal model of the glacier defined by Monaco
& Scozzafava (2015, Fig.6A), in the real dataset inverted model of Fig.5A the boundary conductivity value for the ice rich
layer was set to 1E+1 mS/m, while 2E+2 mS/m represents the ice-debris mixture. These values are two orders of magnitude
higher than those found in the inverted synthetic model (Fig.7A). Note that, this is the same difference exciting between the
instrumental limit resolution (1E-1 mS/m) and the typical electrical conductivity of ice in temperate glaciers (1E-3 mS/m).
Considering all this, we adopted a correction factor of 1E-2 mS/m that has been applied homogeneously to the results of the
inversion process of the field datasets. In this way, as it can be clearly seen in Fig.8, the ice boundaries (ice-rich and ice-debris
mixture) are represented by the same values of the synthetic dataset. The blue dashed line in the FDEM calibrated model shows
the boundary of the ice layer with the underlying bedrock, in very good agreement with the one defined by the GPR model.
The same correction factor has been applied to the inversion results of Line 2 allowing again to define the ice rich layer limit





to of 1E-1 mS/m and ice-debris mixture to 2E-1 mS/m. The calibrated and inverted conductivity section of Line 2 (Fig.9A)
agrees again with the glacier structure defined with the corresponding GPR model and with the synthetic values of fig.7B.
**5 Discussion**
The results of our longitudinal GPR profile (Fig.4A) confirms the negative trend of glacier retreat. In fact, the ice-rich layer
was easily identifiable along the entire GPR profile measured in 2015 by Monaco & Scozzafava, but today seems to end at
x≈30m. This is presumably linked to the loss of massive ice in the last years and increase in the amount of debris. This
interpretation is confirmed by the inverted and calibrated FDEM section (Fig.8A), where the ice-rich layer (σ<1E-1 mS/m)
disappears at x≈30 m. For x<30m the conductivity values are between 1E-1<σ<2E-1 mS/m, suggesting the presence of ice but
probably mixed with considerable quantities of debris. In the GPR profile, the maximum thickness of the ice layer (26.4 m)
can be placed around x≈90 meters. This information agrees with the FDEM section (Fig.8A) where the maximum thickness
of the ice layer seems to be at distance x≈90-100 m. The GPR model highlights a thinning of the ice layer towards the south
direction. On the other hand, in the FDEM model the thickness variation is less evident, confirming the expected lower
resolution of this technique compared to the GPR one. Despite this, the boundary between the ice layer and the bedrock defined
by the FDEM calibrated model (blue line Fig.8A) is very similar to the one defined by the GPR method (blue line Fig.8B).
The goodness of these results is confirmed by the drilling performed in April 2022 by the Ice Memory team. The ice/rock
boundary detected by the drilling was in fact reached at a depth of 27.2 meters from the ground level (ISP-CNR, 2022). It
should be noted that in the calibrated FDEM section (Fig.8A), the layer representing the snow cover with conductivity values
close to 1 mS/m (as defined in the synthetic model of Fig.7A), is missing. This is probably due to the absence of the dataset
acquired in VCP mode and intercoils distance $s = 10$ meters, which involve the shallower layers during the measurements (see
Fig.2). These data configuration has been in fact deleted since we had technical problem with that dataset. On the other hand,
in the GPR model (Fig.4A), the thickness variation of the snow layer moving from south to north is clearly visible (see also
Fig.A1 Appendix). In the southern area, the snow cover is a couple of meters, while towards the glacier front (north) it tends
to increase up to 5 meters (as measured also during the field operations). A similar trend is found also in the GPR profile Line
2 (Fig.4B). The snow layer has a greater thickness in the east direction and thins out moving towards the west (see also Fig.A2
Appendix). In this case, the variation is detected also by the calibrated FDEM section (Fig.9A), where the dataset VCP $s = 10$
m was considered. The GPR profile Line 2 confirms the presence of the ice layer but with a maximum thickness slightly lower
than that found for the longitudinal profile. This is in line with the trend defined by the results of Line 1, where the maximum
thickness of the ice layer is found at x≈90 m, but afterwards it thins out both downstream and upstream. Along the Line 2, the
ice thickness is greater in the center of the profile (50<x<70 m) and tends to thin out both eastwardly and westwardly, as
confirmed also by the calibrated FDEM section (Fig.9A).
It should be noted that all FDEM inverted models have lower penetration depth than those predicted by the instrument
manufacturer (see Fig.2). This is expected, since the investigation depth decreases in subsoils with high electrical resistivity
values (Hauck and Kneisell, 2008). We calculated, for each coil configuration, a sensitivity profile of the measurements related
to the depth (e.g. see Fig A3 Appendix). The inverted FDEM models here presented are limited to the depths where the
normalized sensitivity of the measurements reaches zero, approximately 30 meters in all the profiles.
**6 Conclusions**
The results of the geophysical investigations performed on the Calderone Glacier confirm the excellent capabilities of the GPR
method in glacial environments. The measurements acquired with modern 200 MHz digital antenna define with extreme
precision the thickness of the snowpack and the boundary depth between the ice layer and the calcareous bedrock, a result that
was confirmed by the drilled borehole in April 2022. A future development for the GPR measurements collected on the





Calderone Glacier is to apply the method proposed by Santin et al. (2022), to estimate the debris content within the layer
composed of ice-debris mixture. This method, in the case of periodic measurements performed on the Calderone glacier, can
help to estimate the ice volume losses in the next future.
The results obtained with the separated-coils FDEM device on the Calderone Glacier suggest the potentiality of the induced
electro-magnetic technique, even in a high resistive environment. As in the case of the investigations performed on rock
glaciers by Pavoni et al. (2021), the method does not allow to replicate the real electrical conductivities of the layers which
compose the frozen subsoil, but allows to define the subsoil structure in a relative way. Reproducing the real conductivity
values of the layers with ice was in fact out of the scope, considering the instrumental limit resolution of 1E-1 mS/m. The
results of the FDEM forward modeling, which moreover does not consider the instrumental limit, demonstrate that in these
environments is not possible to find an inverted conductivity section with the real values of the layers, even applying the
Maxwell full solution in the inversion of synthetic datasets (see McLachlan et al., 2021). The inversion result of the FDEM
Line 1 real dataset (Fig.5A), filtered and smoothed considering the non-ideal conditions of the coils during the measurements
(homogeneous distance and orientation), suggest a subsoil structure very similar to the synthetic model (Fig.7A), but with the
conductivity scale two orders of magnitude higher than the expected. This difference of magnitude is the same existing between
the value of the instrumental limit and the conductivity of the ice in a temperate glacier. By simply applying a correction factor
of 1E-2 mS/m to the inversion results of the field dataset, we found an inverted electrical conductivity scale in agreement with
the predicted synthetic models. Therefore, the FDEM surveys on the Calderone Glacier demonstrate once again the importance
of performing the forward modeling process in order to better evaluate the results of the field dataset inversion.
The quality of the data processing applied to the FDEM measurements is confirmed both by the results of GPR surveys and
by the drilling realized at the end of April 2022. The structure found in the inverted and calibrated FDEM model of Line 1 is
practically the one found by the GPR model and by the drilled borehole (see Fig.8). Considering these promising results, the
future project is to use the FDEM separated-coils device in the rock glacier periglacial environments. In these environments
the GPR technique is in fact more complicated to be applied, considering the blocky surface that hinders data acquisition and
enhance the problem of signal scattering. The FDEM method is not affected by these problems and doesn't need galvanic
contact with the blocky surface as the ERT method. Moreover, the logistic effort of the FDEM investigation is much lower if
compared to the ERT survey, therefore it could represent a reliable preliminary investigation to evaluate the subsoil structure.
To conclude, FDEM method should not be proposed as a substitute of the GPR technique in glacier environment, which remain
the best in term of resolution, but rather as a convenient integration able to support the reconstruction of glaciers structure.





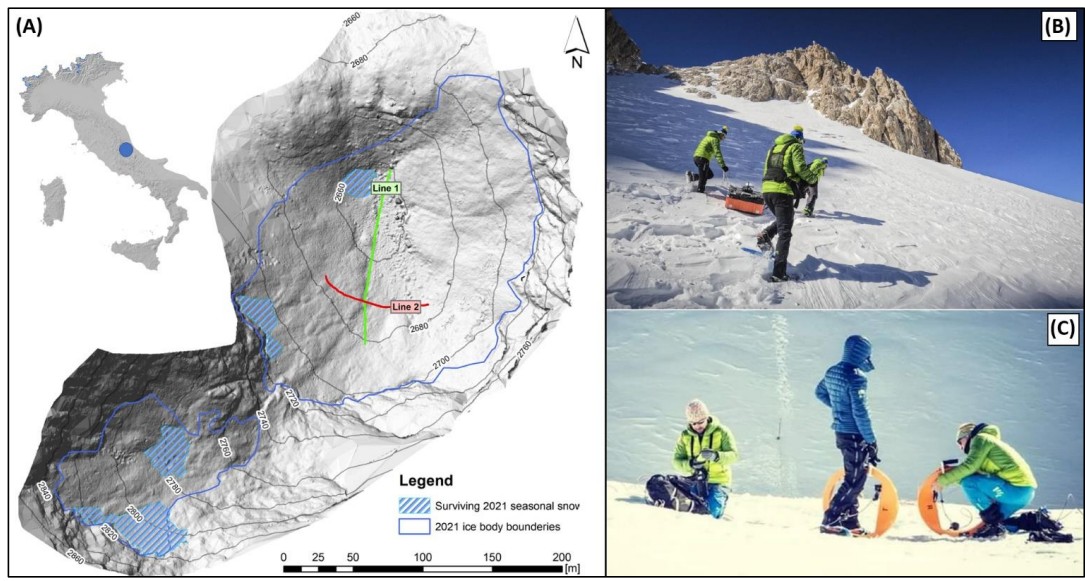

**Figure 1. A) Position of the southernmost glacier of Europe: the Calderone Glacier (blue circle) in Central Italy (EU-DEM v1.1 - Copernicus Land Monitoring Service) and the location of the survey lines performed with the B) GPR and C) FDEM methods. In Fig.1A, the hillshade raster from photogrammetric DTM, survey Line 1 (green line) is 135 meters long and it is longitudinal to the development of Calderone Glacier; Line 2 (red line) is 85 meters long and it is orthogonal to the development of Calderone Glacier.**

| Investigation Range (ns) | Samples (points) | Simple for second | Dynamic (bit) |
|---|---|---|---|
| 400 | 1024 | 40 | 32 |

**Table 1. GPR acquisition parameters used during the measurements performed on the Calderone Glacier survey in March 2022**

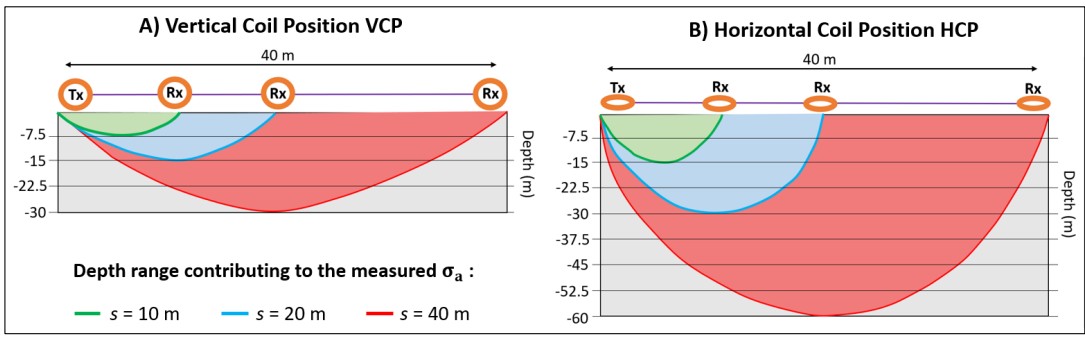

**Figure 2. A) Nominal depth range influencing the measured apparent conductivity $\sigma_a$ for the different CMD-DUO coil separation (s) using the vertical coil orientation (VCP). B) Depth range influencing the measured apparent conductivity $\sigma_a$ for the different CMD-DUO coil separation (s) using the horizontal coil orientation (HCP).**



| | Snow Cover | Frozen Debris | Massive Ice | Calcareous Bedrock |
|---|---|---|---|---|
| Conductivity (mS/m) | 1 | 2E-2 | 1E-3 | 2E-1 |

**Table 2. Electrical conductivity values from literature and used to perform the forward modeling process in the Calderone survey.**

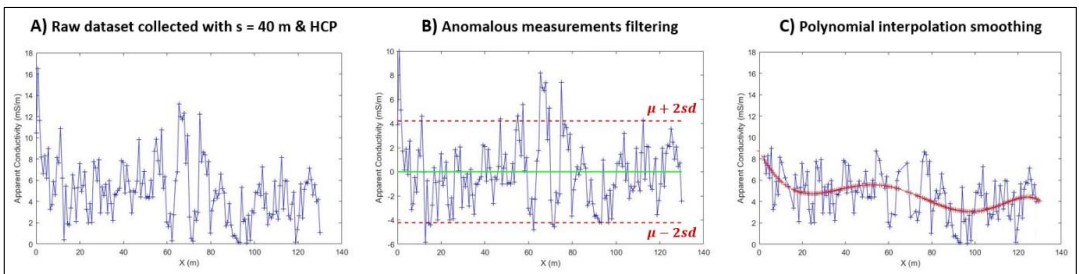

**Figure 3. Example of the data filtering applied to the raw measurements of Line 1. A) Raw dataset acquired with coil separation of 40 meters and horizontal coil orientation. B) After applying a detrend function to the measurements, filtering of the anomalous values which are outside the confidence interval of $\mu - 2sd < \sigma_a < \mu + 2sd$ (μ is the average $\sigma_a$ and sd is the standard deviation of the measurements). C) Smoothing of the saved data using a polynomial function of 6th grade.**

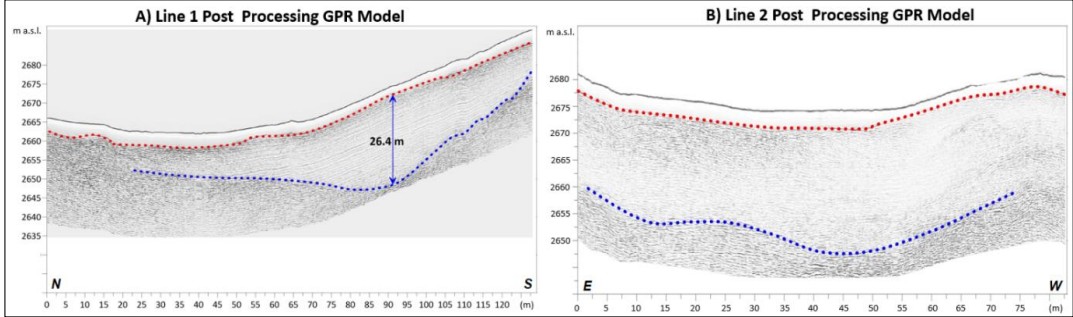

**Figure 4. A) Line 1 and B) Line 2 post-processing GPR models. The red dashed line defines the boundary between the snow cover and the underlying frozen debris. The blue dashed line marks the limit between the ice layer and the bedrock; the blue arrow highlights the maximum thickness of the ice layer found in the Calderone Glacier GPR surveys performed in March 2022. Note that, this is the position where the drilling has been performed in April 2022 and the ice core has been extracted.**



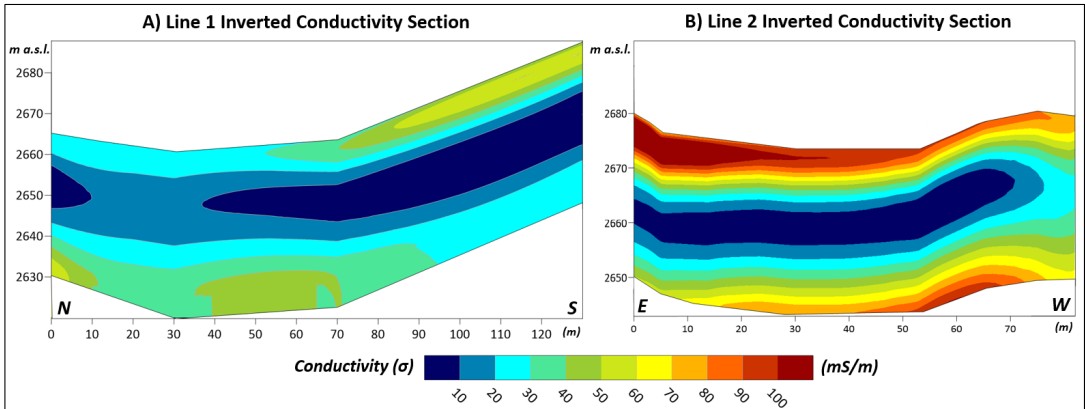

**Figure 5. A)** Inverted Conductivity Model obtained with the dataset collected along Line 1; **B)** Inverted Conductivity Model found with the dataset acquired along Line 2. Note that, in the inversion procedure applied to the longitudinal profile Line 1, the dataset collected with coil separation s = 10 meters and VCP mode has been deleted as particularly noisy.

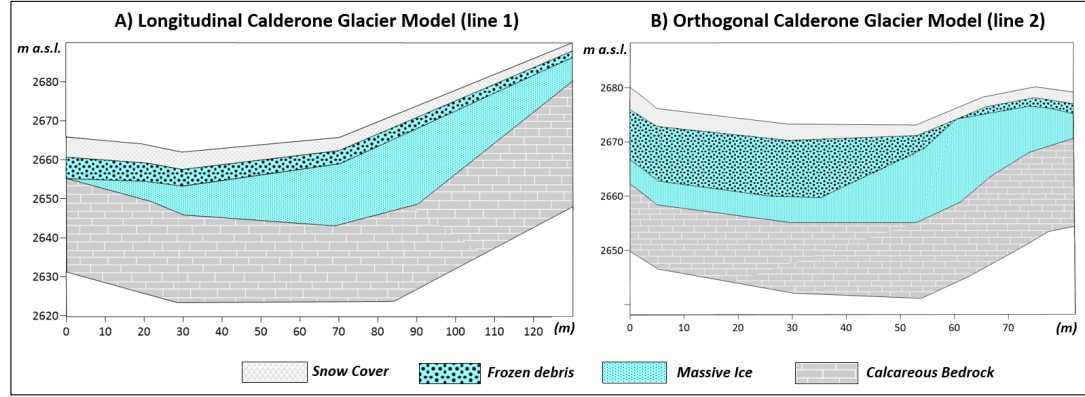

**Figure 6. A)** Longitudinal model of the Calderone Glacier defined by Monaco & Scozzafava (2015). **B)** Orthogonal Calderone Glacier model (below Line 2) defined after the GPR surveys performed in March 2022.

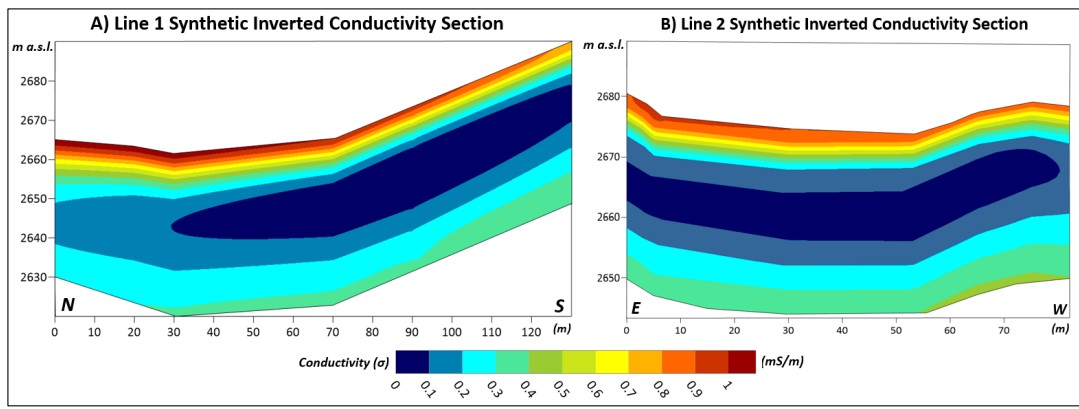

**Figure 7.** Inverted conductivity sections using the synthetic datasets calculated with the forward modeling procedure for the Calderone Glacier models below **A)** Line 1 (Fig.7A) and **B)** Line 2 (Fig.7B).





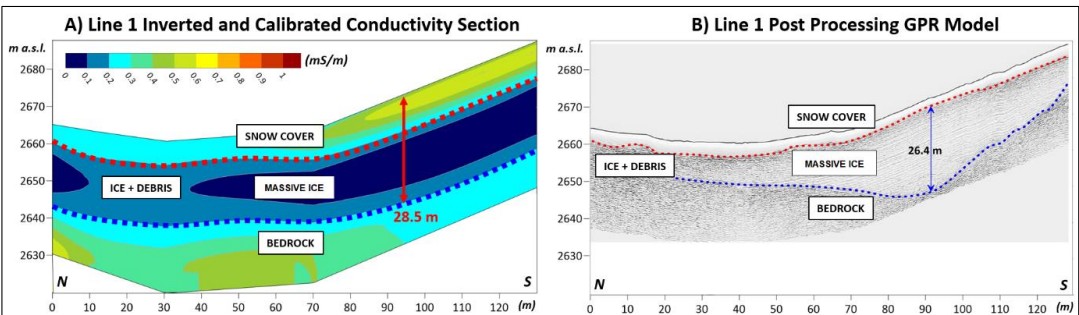

Figure 8. A) Inverted and calibrated conductivity section found applying the correction factor of 1E-2 mS/m to the results of the inversion process of datasets collected in Line 1. The red arrow shows the ice layer boundary in the same location of the B) GPR survey Line 1 (blue arrow). Note that in both the models have been inserted the boundaries between snow cover-frozen debris (red dashed line) and ice layer-bedrock (blue dashed line).

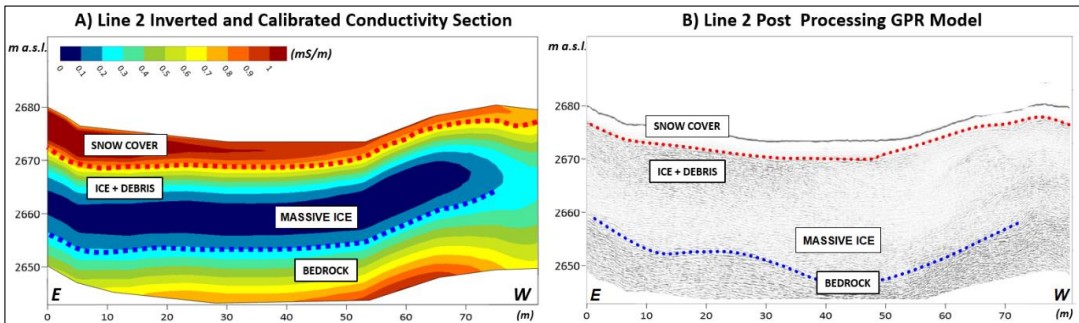

Figure 9. A) Inverted and calibrated conductivity section found applying the correction factor of 1E-2 mS/m to the results of the inversion process of datasets collected in Line 2. B) GPR result of line 2. Note that in both the models have been outlined the boundaries between snow cover-frozen debris (red dashed line) and ice layer-bedrock (blue dashed line).

**Appendix**

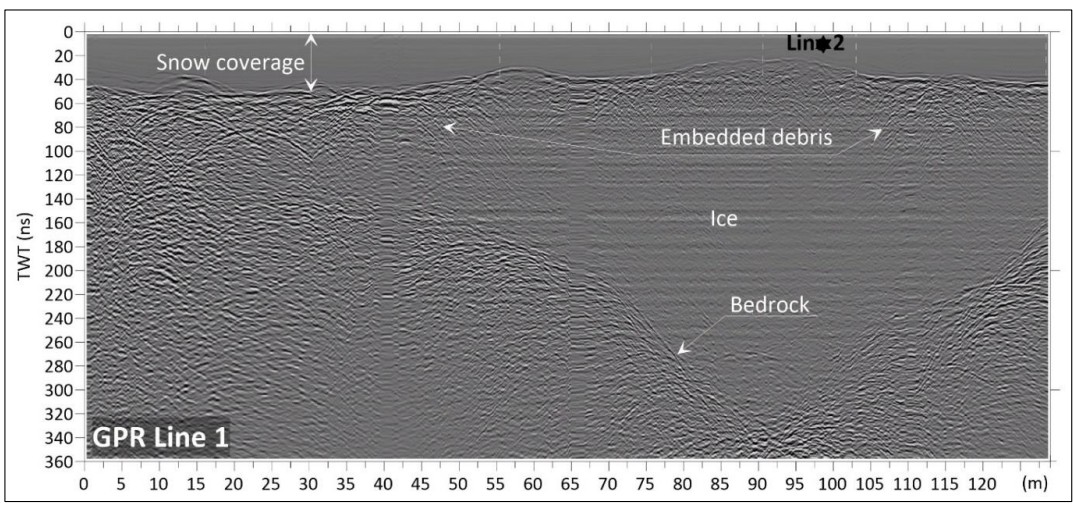

Figure A1. Intepetation of the GPR model Line 1 pre-processing.



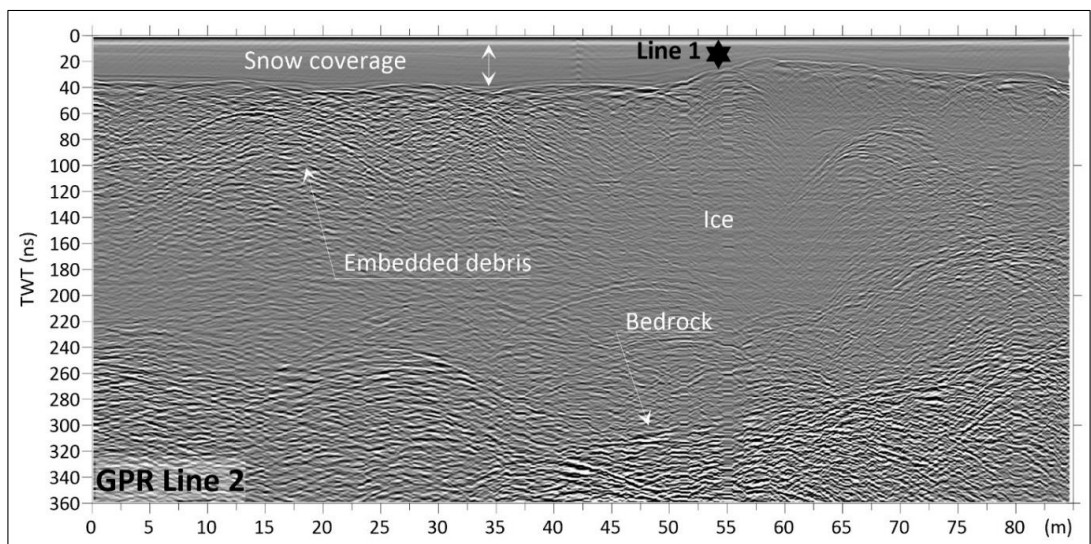

**Figure A2. Intepretation of the GPR model Line 2 pre-processing.**

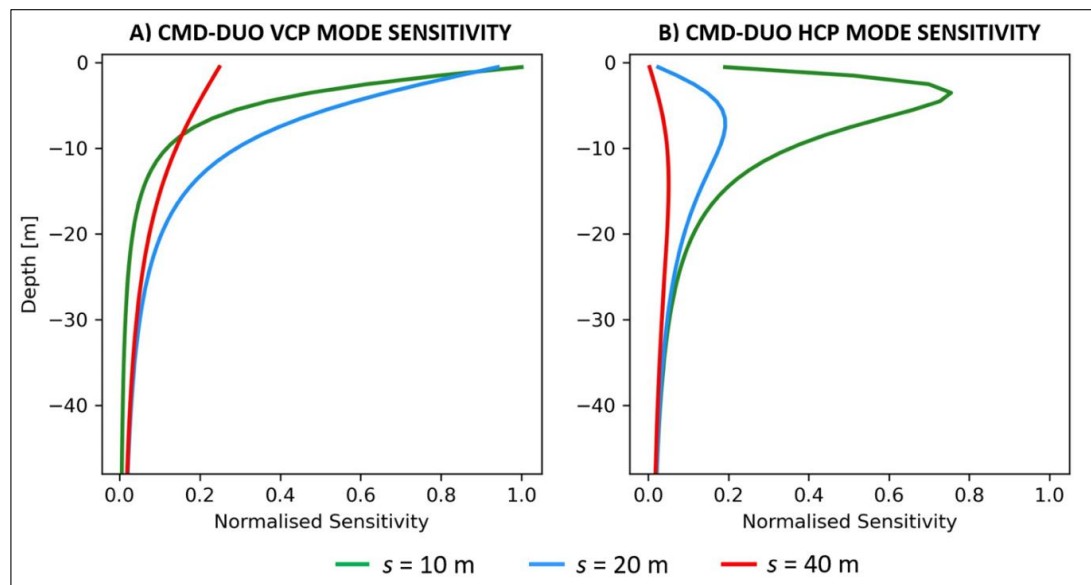

**Figure A3. Normalized sensitivity pattern calculated for the measurements collected along Line 2.**

*Author contributing*. All the authors have been involved in data acquisition; MP performed the data processing of FDEM method; SU performed the data processing of GPR method; all authors contributed to writing and editing the manuscript.

*Acknowledgements*. The Authors thank Massimo Pecci for the relevant discussion about the Calderone Glacier history, the National Fire Department (Corpo Nazionale dei Vigili del Fuoco) for the logistic helicopter support, Pinuccio D'Aquila (Engeoneering Srls) for the photogrammetric acquisition survey of Calderone Glacier (Fig.1A), the photographer Riccardo Selvatico for the images presented in Fig.1B and Fig.1C, and the mountain guides Paolo Conz and Thomas Ballerin for the field support.



*Data Availability Statement.* Datasets used in the current work will be sent to interested researchers upon request.

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
