# Peer review of "Induced Electromagnetic prospecting for the characterization"

_The Cryosphere, 2022_

## Author Comment (AC2)

We respect the comments of the ANONYMOUS reviewer, even if we partially disagree with some of her/his conclusions. Here below a point by point reply.

1) The authors present a multidisciplinary study to investigate the structure of a glacier ice body. They apply the methods of ground penetrating radar and induced electromagnetic prospecting. For the latter technique the authors utilize a CMD-DUO system. This system relies on the principle that an electromagnetic field is emitted by a Tx coil. In a conductive subsurface, secondary currents are induced and the superposition of the primary signal from the transmitter and the secondary signal from the induced eddy currently in the subsurface are recorded in the Rx coil. This principle is inherently limited to applications where the subsurface is sufficiently conductive to induce eddy currents large enough to exceed the detection threshold of the recording system. The manufacturer of the used system gives this value at a limit of 10E-1 mS/m, two orders of magnitude larger than the expected conductivity of massive ice. The authors have explained these limitations correctly in their manuscript.

Consequently, the recorded data show no evidence of sufficiently large induced signals, as evident from Figure 3. Here, the data scatter around zero, large parts of the data showing physically implausible negative values. In a next step "Anomalous measurements filtering" is applied to remove outliers larger than two standard deviations. In the figure 3C where polynomial interpolation smoothing is applied the data appear all positive. This cannot be explained by the presented processing steps.

**Reply 1:** As pointed out by the reviewer, in the manuscript we have widely highlighted the principles of the FDEM method and the instrumental limitation of the CMD-DUO device. We should better underline that the resolution capabilities are linked to the volt-meter installed, and it does not represent the range of application. Therefore, all our interpretations are done in a relative way, as explained.

We must specify that Fig.3 was misinterpreted by the Reviewer, since no negative data values are presented, and we did not record data scattering around zero. Fig.3A shows the raw dataset of Line 1 acquired with coils separation of 40 meters and horizontal coils mode orientation. Figure 3B shows data scattering around zero and negative values after a detrend function was applied to the raw dataset showed in Fig.3A. Detrending (shifting) removes both offsets and linear trends, and was adopted just to remove outliers. For this reason, in Fig.3B negative values and scattering around zero are found (as expected). Once we removed the outliers from the detrended data, the saved measurements have been brought back to their initial raw values and then interpolated with a polynomial function, as shown in Fig.3C. We are sorry if this process was not clear to the Reviewer 2 (on the contrary of Reviewer 1), and we will better explain the filtering steps in the revised version of the manuscript. Obviously, all the raw data will be available for TC readers in case they intend to replicate the processing.

It seems that most of the anonymous Reviewer criticisms are based on the assumption that EMI methods cannot be applied in high resistive environments. It is true that few eddy currents can be induced in low conductive layers but, as underlined in chapter 3.2 and in Fig.2, the measured apparent conductivities derive from the contribution of the whole deposits that compose the subsurface. As we can see in Fig.3C, higher apparent conductivities are measured for x<40 m and lower values are found for x>40 m, suggesting that the induction of eddy currents is facilitated in the first part of the transect, where in fact the ice layer has lower thickness (see Fig.6A). On the other hand, for x>40 m, the apparent conductivity values decrease since the thickness of the ice layer increase and hinders the induction of large amount of eddy currents in the subsurface. The structure of ice layers explaining our relative interpretation is confirmed by independent GPR and borehole measurements. The instrument resolution limit is in fact not linked to the induction capabilities (as the Reviewer seems to assert), but to the ability of the instrument to detect the weak low voltage received. Even if generating eddy currents in low conductive environments is challenging, EMI methods (regardless in time or frequency domains) have been historically applied in glacial and periglacial environments with ice rich layers in the subsurface, e.g.:

- Bucki, A., Echelmeyer, K., & MacInnes, S. (2004). The thickness and internal structure of Fireweed rock glacier, Alaska, U.S.A., as determined by geophysical methods. Journal of Glaciology, 50(168), 67-75.
- Cockx, Liesbet, et al. Prospecting frost-wedge pseudomorphs and their polygonal network using the electromagnetic induction sensor EM38DD. Permafrost and Periglacial Processes 17.2 (2006): 163-168.
- Daniel Blatter, Kerry Key, Anandaroop Ray, Neil Foley, Slawek Tulaczyk, Esben Auken, Trans-dimensional Bayesian inversion of airborne transient EM data from Taylor Glacier, Antarctica, Geophysical Journal International, Volume 214, Issue 3, September 2018, Pages 1919–1936.
- Foged, N., et al. "Airborne and ground-based TEM mapping in polar regions—Antarctica cases. NSG2021 2nd Conference on Geophysics for Infrastructure Planning, Monitoring and BIM. Vol. 2021. No. 1. European Association of Geoscientists & Engineers, 2021.
- Harada, K., Wada, K. and Fukuda, M. (2000). Permafrost mapping by transient electromagnetic method. Permafrost and Periglacial Processes, 11, 71–84.
- Harada, K., Wada, K. and Fukuda, M. (2003). Detection of permafrost structure by transient electromagnetic method in Mongolia. Proceedings of the 8th International Conference on Permafrost, Zurich, Switzerland, Extended Abstracts Reporting Current Research and New Information, 53–54.
- Hauck, C., Guglielmin, M., Isaksen, K. and Vonder Muhll, D. 2001. Applicability of frequency domain and time domain electromagnetic methods. Permafrost Periglac. Process, 12(1), 39–52.
- Hauck, C., Mühll, D.V. (1999). Detecting alpine permafrost using electro-magnetic methods. In: Hutter, K., Wang, Y., Beer, H. (eds) Advances in Cold-Region Thermal Engineering and Sciences. Lecture Notes in Physics, vol 533. Springer, Berlin, Heidelberg.
- Hoekstra, Pieter, Paul V. Sellmann, and Al Delaney. Ground and airborne resistivity surveys of permafrost near Fairbanks, Alaska. Geophysics 40.4 (1975): 641-656.
- Grombacher, D., Auken, E., Foged, N., Bording, T., Foley, N., Doran, P. T., ... & Tulaczyk, S. (2021). Induced polarization effects in airborne transient electromagnetic data collected in the McMurdo Dry Valleys, Antarctica. Geophysical Journal International, 226(3), 1574-1583.
- Madsen, L. M., Bording, T., Grombacher, D., Foged, N., Foley, N., Dugan, H. A., ... & Auken, E. (2022). Comparison of ground-based and airborne transient electromagnetic methods for mapping glacial and permafrost environments: Cases from McMurdo Dry Valleys, Antarctica. Cold Regions Science and Technology, 199, 103578.
- Maurer, H., & Hauck, C. (2007). Geophysical imaging of alpine rock glaciers. Journal of Glaciology, 53(180), 110-120.
- Neil Foley, Slawek Tulaczyk, Esben Auken, Cyril Schamper, Hilary Dugan, Jill Mikucki, Ross Virginia, Peter Doran; Helicopter-borne transient electromagnetics in high-latitude environments: An application in the McMurdo Dry Valleys, Antarctica. Geophysics 2015; 81 (1): WA87–WA99.
- Petersen, E., Holt, J., Stuurman, C., Levy, J. S., Nerozzi, S., Paine, J. G., ... & Fahnestock, M. (2016, March). Sourdough Rock Glacier, Alaska: An analog to martian debris-covered glaciers. In 47th Lunar and Planetary Science Conference (Vol. 2535).

2) After applying an inverse modelling, the results do not match plausible values for glacier ice bodies. This can logically explain by improper input data. The authors confirm the implausible data and apply an empiric shift of two orders of magnitude, justified by the misfit between synthetic data and FDEM models from field data. No physical explanation for this approach is presented. This empiric shift of the data is by no means a "calibration", it is rather an adaptation to the expected values.

**Reply 2**: We agree that we wrong term speaking about "calibration" and we thank the anonymous Reviewer for this comment. The term "correction" is more suitable, since we introduced just a fixed shifting correction factor (1E-2 mS/m). The latter has been defined by comparing the inverted models from the field datasets

(Fig.5) and the ones obtained from the synthetic datasets (Fig.7). Considering the instrumental limit resolution, from the inversion of field datasets we cannot expect to find conductivity values in the same range of table 2. Therefore, synthetic forward modelling process was computed to verify the obtained results. The fixed correction factor is applied to the inversion results of the field datasets and allows to have models with the same conductivity range as in the inverted synthetic models (from 0 to 1 mS/m – see Fig.7). In this way, as you can see in Fig.8 and Fig.9 (FDEM inverted and corrected sections), the range of conductivity in the field dataset models span from 0 to 1 mS/m, and the ice layer boundaries can be defined with the same values found in the synthetic models (0-0.1 for the ice rich layer and 0.1-0.2 for the ice-debris mixture, see chapter 4.3). We are asserting that real data fit the synthetic ones, suggesting the same subsoil structure with conductivities just shifted to higher values. Note that, we applied the same "correction factor" procedure to the result of FDEM surveys performed in rock glacier environments, and the resulting subsoil structures were confirmed by the ERT surveys carried out on the same investigation lines.

3) It is common practice for models obtained by inversion techniques to present i) the full recorded data, ii) the derived model, iii) the synthetic data, predicted by the model and iv) residuals of modelled and observed data. Ideally together with information about the performance/convergence of the inversion and the final data misfit e.g. as RMS. Based on this information, the reliability of the model can be evaluated. Such information would be mandatory for the manuscript. According to journal standards, the full dataset should be made available as supplementary data, ideally together with the data processing algorithm, to give the reader transparent insight in the robustness of the approach.

**Reply 3:** We will insert in chapter 4.2 (FDEM inversion results) the RRMSE (Relative Root Mean Squared Error) to evaluate the accuracy of both the inverted models (line 1 and line 2) and we will create a public GitHub repository where we will insert FDEM and GPR datasets (not possible in the pre-print version). As we highlighted in the manuscript, EMagPy is a published Python-based open-source software to process FDEM forward and inverse modelling. Therefore, there is no algorithm that we need to share since it is already easily and freely avaible to download (see McLachlan et al., 2021).

4) The derived model in Figures 7&8 do not show any convincing correlation to the internal structure of the investigated ice body, the derived conductivities largely follow the topography and model boundaries. In Figure 8, the authors interpret the internal structures arbitrarily at the colour contours of 0.1, 0.2, 0.3, 0.4 mS/m without any physical reasoning. In the introduction section the authors have introduced such units with conductivities orders of magnitude different.

**Reply 4:** We respect the comment of the anonymous Reviewer, but we do not agree with her/his opinion. Our FDEM sections define a subsurface structure very similar to the GPR models, collected independently by another research group, as we highlighted in Fig.8 and Fig.9. To confirm our findings (e.g. the ice layer boundaries) we performed the forward modelling process (see chapter 4.3 FDEM forward modelling results). As we underlined in the conclusion chapter of the manuscript, the results of the FDEM forward modelling, which does not consider any instrumental limit, demonstrate that in these low conductive environments it is not possible to retrieve the real conductivity values of the layers (compare values of table 2 and Fig.7), even applying the Maxwell full solution in the inversion of synthetic datasets, however we can retrieve the subsoil structure. Consequently, a low conductive environment does not exclude the use of EMI methods if the results are interpreted in a relative way to define the subsurface structure. In our case, the interpretation of the ice layer boundaries (<0.1 mS/m ice rich layer, 0.1-0.2 mS/m ice and debris) have been defined by the synthetic modelling (Fig 7), while subsoil geometries were confirmed from independent GPR data.

The model boundaries tend to follow the topography, both for FDEM and GPR models, since the glacier itself influences the morphology of the surface. Obviously, in the GPR model the thickness variations of the ice layer are more appreciable, as expected given the higher resolution of the method. FDEM technique has clearly less capability to detect these lateral variations since it is a 1D vertical survey, and the CMD-DUO

device has a very large coil separation (10-20-40 m). Moreover, the FDEM data inversion is still 1D, this means that the results are vertical conductivity profiles that can be interpolated in pseudo-2D conductivity section, as discussed in this work. Moreover, FDEM datasets have been acquired in challenging condition and it was complicated to guarantee the perfect coils orientation, height, and separation during the measurements. For these reasons we applied to the datasets the smoothing presented in Fig.3C, which may have hidden some information as small lateral conductivity variations.

5) I understand that the authors have invested a substantial field effort to acquire and process this dataset and compile this manuscript. But on the basis of i) the inconsistent approach in applying the technique of FDEM to an environment of such highly resistive ground, ii) the shortcomings in the data processing, iii) the incomplete information of the inversion and iv) the inconsistent interpretation, I cannot recommend this study for publication in The Cryosphere. I see the limitations of the manuscript so fundamental, that even major revisions will not bring the manuscript to a publishable standard and thus recommend rejection, but leave the final verdict to the editor.

**Reply 5**: We respect the opinion of the anonymous reviewer, but we do not agree with her/his conclusions that seem based more on a prejudice respect the EMI methods than on a careful reading of our case study. i) Induction methods can be used in periglacial and glacial environments (see literature above); ii) we adopted a solid inversion process already published in several papers; iii) our findings are confirmed by the comparison with GPR and borehole measurements.  In the corrected manuscript we will better specify the data filtering and processing, as the Reviewer suggested. We hope that the Editor can appreciate the revised version of the manuscript, far from any methodological prejudice, since we think it represents an interesting application for the cryosphere community.